# Fixed-Dose Factor Eight Inhibitor Bypassing Activity (FEIBA) in the Management of Warfarin-Associated Coagulopathies

**DOI:** 10.3390/pharmacy10030050

**Published:** 2022-04-23

**Authors:** Francisco Ibarra, Mallory Cruz, Matthew Ford, Meng-Jou Wu

**Affiliations:** 1Community Regional Medical Center, Fresno, CA 93721, USA; mcruz2@communitymedical.org (M.C.); mwu@communitymedical.org (M.-J.W.); 2Clovis Community Medical Center, Clovis, CA 93611, USA; mford@communitymedical.org

**Keywords:** FEIBA, warfarin, reversal, bleeding

## Abstract

This retrospective review evaluated our institutions’ practice of administering low fixed-dose FEIBA (high (1000 units) or low dose (500 units) for an INR ≥ 5 or <5, respectively) for the management of warfarin-associated coagulopathies. The primary outcome was the percentage of patients who had a post-FEIBA INR ≤ 1.5. In the total population, 55.6% (10/18) of patients achieved a post-FEIBA INR ≤ 1.5. In the subgroup analysis, significantly more patients in the low dose FEIBA group achieved a post-FEIBA INR ≤ 1.5 compared to the high dose FEIBA group (71.4% vs. 45.5%, respectively, *p* < 0.001). In the post hoc analysis, there was a significant difference in the number of patients who achieved a post-FEIBA INR ≤ 1.5 when comparing those who received high dose FEIBA with a baseline INR 5–9.9 to those who received high dose FEIBA with a baseline INR ≥ 10 (60% vs. 33.3%, respectively, *p* < 0.001). The existing literature and our findings suggest that patients who present with lower baseline INR values and receive additional reversal agents are more likely to meet post-reversal INR goals. Current low fixed-dose protocols may be oversimplified and may need to be revised to provide larger fixed-doses.

## 1. Introduction

Warfarin is an oral anticoagulant used in the primary and secondary prevention of arterial and venous thromboembolisms. Warfarin inhibits the vitamin K epoxide reductase enzyme complex and synthesis of vitamin K-dependent clotting factors II, VII, IX, X, and Proteins C and S [1]. A serious adverse effect of warfarin therapy is bleeding, which in some cases may be fatal. In the management of warfarin-associated coagulopathies, vitamin K is administered to allow for the synthesis of vitamin K-dependent clotting factors, but due to its delayed onset of action, prothrombin complex concentrates (PCC) are administered with vitamin K [2]. Prothrombin complex concentrates contain vitamin K-dependent clotting factors and rapidly normalize the international normalized ratio (INR) and restore hemostasis [3,4,5,6].

Kcentra is the only FDA-approved four factor PCC for managing warfarin-associated coagulopathies and is recommended by several national societies [3,4,5,6]. Traditionally, Kcentra is dosed based on the patient’s weight and INR but is also given as a low fixed-dose [7,8,9,10,11,12,13,14,15,16,17]. A low fixed-dose approach offers several advantages over traditional dosing, including decreased time to administration and costs. Although not FDA approved for managing warfarin-associated coagulopathies, Factor VIII inhibitor bypass activity (FEIBA) is used for this indication [18,19,20,21,22]. FEIBA is approved for managing hemophiliac-related hemorrhages and, as opposed to Kcentra, contains activated factor VII.

Despite the positive results of several studies, FEIBA is not formally recognized by national societies as a therapeutic option for managing warfarin-associated coagulopathies [7,8,9,10,11,12,13,14,15,16,17,18,19,20,21,22]. This finding may be driven by FEIBA lacking an FDA approval for this indication, the relatively small number of studies evaluating the use of FEIBA in the management of warfarin-associated coagulopathies compared to PCC, and the preconceived notion that FEIBA is more prothrombotic than PCC due to it containing activated factor VII. However, recently published studies do not support the latter belief [14,15,23,24]. Therefore, our study aimed to evaluate our institutions’ practice of administering low fixed-dose FEIBA for the management of warfarin-associated coagulopathies and contribute to the existing literature.

## 2. Materials and Methods

This was a retrospective study conducted at two institutions within the Community Medical Centers’ health care system. The larger of the two study sites is a 685-bed academic-affiliated medical center with more than 110,000 Emergency Department visits annually, while the smaller site is a 208-bed community hospital. The local Institutional Review Board approved the study protocol. Patients were included if they were 18 years or older and received FEIBA, the institutions’ only formulary four factor PCC, for the management of a warfarin-associated coagulopathy with a baseline INR > 1.5. Patients were excluded if they were pregnant or incarcerated, did not have a post-FEIBA INR, received FEIBA for the reversal of warfarin prior to surgery in the setting of a non-active bleed, or received additional reversal agents (i.e., fresh frozen plasma (FFP), cryoprecipitate, Factor VII) before the first post-FEIBA INR was obtained.

Per our institutions’ Anticoagulation Committee recommendation, patients presenting with a warfarin-associated coagulopathy received high (1000 units) or low dose (500 units) FEIBA intravenously (IV) if their INR was ≥5 or <5, respectively. If the post-FEIBA INR drawn after 30 min was >1.5, patients could receive an additional 500 units of FEIBA at the prescriber’s discretion. In addition to FEIBA, all patients were recommended to receive IV vitamin K 10 mg. Complete adherence to the recommendation was not mandated and prescribers could order additional reversal agents.

The primary outcome was the percentage of patients who had a post-FEIBA INR ≤ 1.5. To evaluate the effectiveness of a single dose, only the first post-FEIBA INR was evaluated. Secondary outcomes included in-hospital mortality and thromboembolic event rates. A subgroup analysis compared the two dosing groups and the pre- and post-FEIBA INR values of those who did and did not achieve a post-FEIBA INR ≤ 1.5. Descriptive statistics were used to summarize patient demographics and outcomes. Continuous data was assessed using the Mann–Whitney U test and nominal data was assessed using the Pearson’s chi-squared test. Statistical significance was determined with a *p*-value < 0.05.

All patients who received FEIBA during 2012–2020 were identified via a system-generated report and screened for inclusion criteria. Demographic and clinical data were collected using a standardized data collection tool by all the study investigators with a Cohen’s kappa coefficient of 0.9. Prior to admission medications, past medical history, warfarin indication, hemorrhage type, in-hospital thromboembolic events (including myocardial infarction, stroke, transient ischemic attack, deep vein thrombosis, and pulmonary embolism), hospital length of stay, and in-hospital mortality were obtained from physician progress notes and discharge summaries. Patients were observed for thromboembolic events throughout their hospital stay only. Baseline and post-FEIBA INR values were obtained from the laboratory results section within the patient’s medical record. Our institutions’ laboratory reports any INR value > 20 as >20. When reporting median baseline INR values, a value of 20 was used for patients with INR values > 20. The medication administration record was reviewed to determine medication doses, administration times, and the use of additional medications. Patients’ actual body weight was used to determine the weight-based FEIBA dose received (units/kg).

## 3. Results

Of the 47 patients screened, 29 were excluded. Patients were excluded for the following reasons: received additional reversal agents (*n* = 13), received FEIBA for the reversal of warfarin prior to surgery in the setting of a non-active bleed (*n* = 9), and did not have a post-FEIBA INR (*n* = 7). Most patients were on warfarin for stroke prevention in non-valvular atrial fibrillation and treatment of venous thromboembolisms (Table 1). The most common hemorrhages were intracranial (*n* = 8) and gastrointestinal (*n* = 6). The median [interquartile range (IQR)] time to drawing the post-FEIBA INR was 121 (61–187) min.

Seven patients presented with an INR < 5 and received low dose FEIBA, whereas 11 patients presented with an INR ≥ 5 and received high dose FEIBA (Table 2). In the total population, 55.6% of patients achieved a post-FEIBA INR ≤ 1.5. The median (IQR) pre- and post-FEIBA INR in the total population was 5.2 (2.9–9.9) and 1.5 (1.4–2.1), respectively. One (5%) thromboembolic event was observed in a patient who received high dose FEIBA and had a history of Factor V Leiden thrombophilia. The event occurred four days after the administration of FEIBA and prior to resuming anticoagulation. Two deaths were observed in the total population, and both were attributed to respiratory failure.

In the subgroup analysis, significantly more patients in the low dose FEIBA group achieved a post-FEIBA INR ≤ 1.5 compared to the high dose FEIBA group (71.4% vs. 45.5%, respectively, *p* < 0.001, Table 3). The pre-FEIBA median (IQR) INR values in the patients who did and did not achieve a post-FEIBA INR ≤ 1.5 were 3.90 (2.75–7.45) and 7.80 (3.20–14.3), respectively (*p* = 0.21). The post-FEIBA median (IQR) INR values in the patients who did and did not achieve a post-FEIBA INR ≤ 1.5 were 1.4 (1.3–1.5) and 2.2 (1.9–3.1), respectively (*p* < 0.001). A post hoc analysis was performed to identify the impact of baseline INR values ≥ 10 on post-FEIBA INR values. In the post hoc analysis, there was no significant difference in the number of patients who achieved a post-FEIBA INR ≤ 1.5 when comparing those who received low dose FEIBA with a baseline INR < 5 to those who received high dose FEIBA with a baseline INR of 5–9.9; 71.4% vs. 60%, respectively, *p* = 0.10. There was a significant difference in the number of patients who achieved a post-FEIBA INR ≤ 1.5 when comparing those who received high dose FEIBA with a baseline INR 5–9.9 to those who received high dose FEIBA with a baseline INR ≥ 10: 60% vs. 33.3%, respectively, *p* < 0.001.

## 4. Discussion

Although direct oral anticoagulants have largely replaced the use of warfarin, its continued use remains a significant cause of morbidity and mortality [25,26,27]. Our study aimed to positively contribute to the existing literature supporting the use of FEIBA in the management of warfarin-associated coagulopathies but determined our institutions’ low fixed-dose FEIBA protocol to be suboptimal. Our findings suggest that current low fixed-dose regimens may be oversimplified and should be re-evaluated.

During warfarin treatment, there is a dose-dependent acquired deficiency of vitamin K-dependent clotting factors. As demonstrated in the clinical studies leading to Kcentra’s approval, larger PCC doses are needed to restore hemostasis in patients presenting with higher degrees of coagulopathy [28]. The baseline INR in our total population was 5.2 (2.9–9.6), and to our knowledge this is higher than all other studies to date evaluating a fixed-dose approach (Table 4). Five studies have evaluated our reversal approach and three reported higher efficacy rates than us, but their baseline INR values were ≤4 [19,21,23]. In seven studies which evaluated a larger fixed-dose approach than us, six reported higher success rates and all had lower baseline INR values [7,9,10,13,14,20]. In our study, the baseline INR of patients who achieved a post-FEIBA INR ≤ 1.5 was half that of those who did not meet this outcome and their post-FEIBA INR was significantly less. The statistical significance observed when comparing the number of patients who achieved a post-FEIBA INR ≤ 1.5 following the administration of low and high dose FEIBA was absent when the high dose FEIBA group was revised in the post hoc analysis to only include patients with baseline INR values of 5–9.9. Our findings suggest that patients with lower baseline INR values are more likely to achieve target post-reversal INR values and that current low fixed-dose protocols may need to be adjusted to account for higher baseline INR values.

Differences in success rates between the other studies and ours may further be explained by the exclusion of patients who received additional reversal agents in our study. In six studies, FFP was co-administered to >20% of the population and the rates of success were >70% [7,9,10,15,19,20]. Although FFP contains fewer clotting factors than PCC, the additional clotting factors provided by FFP are anticipated to enhance the hemostatic effectiveness of PCC. As such, the hemostatic effectiveness associated with combining FFP with PCC may be comparable to increasing the PCC dose. However, PCC are recommended over FFP, and the latter approach may be preferred [4,5]. These findings support the notion that larger PCC doses result in higher success rates and current low fixed-dose protocols may need to be modified.

Many fixed-dose regimens recommend a second PCC dose if the INR remains elevated following administration of the first dose. Although PCC can be prepared and administered more quickly than FFP, failure to rapidly correct the INR is associated with poor outcomes [4]. Institutions with emergency medicine pharmacy services may reduce the time to PCC administration, but institutions like ourselves that lack 24/7 emergency medicine pharmacy services are subject to further delays in administration [29]. Therefore, the initial PCC dose administered should be sufficient to restore hemostasis.

Although our study did not demonstrate a high success rate, our findings do not exclude the use of low fixed-dose regimens in the management of warfarin-associated coagulopathies. Since warfarin does not inhibit exogenously administered clotting factors, lower PCC doses can be utilized, but the ideal low fixed-dose regimen has yet to be determined. The 2020 American College of Cardiology (ACC) guidelines recommend, as an alternative to traditional Kcentra dosing, low fixed-doses of 1500 and 1000 units for patients presenting with and without an intracranial hemorrhage, respectively, irrespective of the INR [5]. In the Gilbert et al. study, this alternative regimen resulted in 86.7% of the total population achieving an INR ≤ 1.5 [15]. When compared to traditional dosing (90%), no significant difference was found (*p* = 0.68). However, the median baseline INR values were ~3 and 23.3% of the total population received FFP. Bitonti et al. evaluated a similar regimen but failed to demonstrate non-inferiority in the number of patients who achieved an INR < 1.5 when comparing their fixed-dose regimen (1500 units for all patients unless the INR > 7.5 or patient’s weight > 100 kg then they received 2000 units) to traditional dosing (75% vs. 90%, respectively) [14]. In comparison to the Gilbert et al. study, the baseline INR values were higher in the Bitonti et al. study, and ≤10% of the total population received FFP. Lastly, Rowe et al. compared our fixed-dose regimen to traditional dosing in a population with an average baseline INR of 2.7 and found no significant difference in the number of patients who achieved an INR < 1.4 (77.1% vs. 62.5%, respectively, *p* = 0.075) [23]. These findings further demonstrate the influence of baseline INR values and co-administration of additional reversal agents on outcomes.

Our study has several limitations, most notably our small sample size. To account for the replacement of warfarin by direct oral anticoagulants, we broadened our study time period, but were still limited in the number of patients we included. We defined efficacy by a surrogate marker rather than clinical achievement of hemostasis, which could have influenced our success rates. However, our endpoint is aligned with previously published studies. Lastly, due to significant differences in study design, interventions, and population characteristics, direct comparisons between our study and those already published cannot be made and our observations should be further investigated.

## 5. Conclusions

The use of a low fixed-dose PCC regimen for the management of warfarin-associated coagulopathies in our study was effective in approximately half of the population and lower than other studies. During warfarin treatment, there is a dose-dependent acquired deficiency of vitamin K-dependent clotting factors, and larger PCC doses are needed to restore hemostasis in patients presenting with higher degrees of coagulopathy. The lower rates of success observed in our study may be attributed to the inclusion of patients with higher baseline INR values and exclusion of patients who received additional reversal agents. The existing literature and our findings suggest that patients who present with lower baseline INR values and receive additional reversal agents are more likely to meet post-PCC INR goals. This study does not exclude the use of low fixed-dose regimens in the management of warfarin-associated coagulopathies but suggests that current protocols are over simplified and need to be revised. Our findings will be used to revise our current low fixed-dose protocol.

## Figures and Tables

**Table 1 pharmacy-10-00050-t001:** Baseline values.

Total Population, *n*	18
Male, *n* (%)	13 (72)
Age—y, mean (standard deviation)	69 (12)
Weight—kg, mean (standard deviation)	87 (33)
Laboratory values, mean (standard deviation)	
Alanine transaminase, U/L	25 (45)
Aspartate aminotransferase, U/L	44 (73)
Creatinine, mg/dL	1.5 (1.1)
Hemoglobin, g/dL	10.5 (3.6)
Warfarin indication, *n* (%)	
Atrial fibrillation (non-valvular)	7 (39)
Atrial fibrillation (valvular)	2 (11)
Thromboembolism	7 (39)
Valve replacement	2 (11)
Past medical history, *n* (%) ^†^	
Cancer	1 (5.6)
Cerebral vascular accident	8 (44.4)
Diabetes	6 (33.3)
Heart failure	7 (38.9)
Hypertension	12 (66.7)
Myocardial infarction	2 (11.1)
Antiplatelet use, *n* (%)	9 (50)
Hemorrhage location, *n* (%)	
Hemothorax	1 (5.6)
Intracranial	8 (44.4)
Gastrointestinal	6 (33.3)
Genitourinary	1 (5.6)
Musculoskeletal	1 (5.6)
Pericardial effusion	1 (5.6)
Traumatic hemorrhage, *n* (%)	6 (33.3)

^†^ several patients had more than one past medical history.

**Table 2 pharmacy-10-00050-t002:** Results.

FEIBA Dose Groups, *n* (%)	
Low (INR < 5)	7 (38.9)
High (INR ≥ 5)	11 (61.1)
Pre-FEIBA INR values—median (interquartile range)	
Total population	5.2 (2.9–9.6)
Low dose	2.9 (2.6–3.4)
High dose	9.2 (6.0–18.1)
Post-FEIBA INR values—median (interquartile range)	
Total population	1.5 (1.4–2.1)
Low dose	1.5 (1.3–1.7)
High dose	1.9 (1.4–2.4)
Post-FEIBA INR ≤ 1.5, *n* (%)	
Total population	10 (55.6)
Low dose	5 (71.4)
High dose	5 (45.5)
FEIBA dose, units—median (interquartile range)	
Total population	923 (574–1044)
Low dose	574 (528–574)
High dose	1016 (923–1062)
FEIBA dose, units/kg—median (interquartile range)	
Total population	8.3 (7.4–13.8)
Low dose	8.0 (7.2–8.5)
High dose	11.7 (7.5–14.6)
Thromboembolic rate, *n* (%)	1 (5.6)
Length of stay—d, median (interquartile range)	8.2 (7.0–11.3)
Survived, *n* (%)	16 (88.9)

**Table 3 pharmacy-10-00050-t003:** Subgroup and post hoc analysis of patients who achieved a post-FEIBA INR ≤ 1.5.

	Yes	No	*p*-Value
N (%) ^†^	10 (55.6)	8 (44.4)	-
Low dose	5 (71.4)	2 (28.6)	-
High dose	5 (45.5)	6 (55.5)	-
INR values—median (interquartile range)			
Pre-FEIBA	3.90 (2.75–7.45)	7.80 (3.20–14.3)	0.21
Post-FEIBA	1.4 (1.3–1.5)	2.2 (1.9–3.1)	<0.001
FEIBA dose, units/kg—median (interquartile range)	9.3 (7.1–15)	8.1 (7.5–11)	0.56
Post hoc analysis with revised baseline INR values, *n* (%) ^††^			
<5	5 (71.4)	2 (28.6)	-
5–9.9	3 (60)	2 (40)	-
≥10	2 (33.3)	4 (66.7)	-

^†^ Low vs. high: *p* < 0.001. ^††^ <5 vs. 5–9.9: *p* = 0.10; 5–9.9 vs. ≥10: *p* < 0.001.

**Table 4 pharmacy-10-00050-t004:** Comparison of other studies.

Study	Agent	Dosing	Baseline INR	INR Goal	% Goal Met
This study	aPCC	INR < 5:500 INR ≥ 5:1000	5.2 (2.9–9.6)	≤1.5	55.6
Wojcik [18]	aPCC	INR < 5:500 INR ≥ 5:1000	3.3 (1.2–∞)	≤1.4	50.7
Stewart [19]	aPCC	INR < 5:500 INR ≥ 5:1000	3.56 (1.3–6.80)	≤1.5	88
Htet [21]	aPCC	INR < 5:500 INR ≥ 5:1000	4 (2.7–7.3)	≤1.5	93
Rowe [23]	aPCC	INR < 5:500 INR ≥ 5:1000	2.7 (1.9)	<1.4	77.1
Dietrich [24]	aPCC	INR < 5:500 INR ≥ 5:1000	2.9 (2.4–4.4)	≤1.4	52.4
Klein [7]	PCC	1500	3.3 (2.5–4)	≤1.5	71.8
Scott [9]	PCC	1000	2.84 (1.18)	≤1.4	73
Varga [13]	PCC	1000	2.8 (2.2–3.4)	≤1.5	48.5
Carothers [20]	aPCC	1000	2.6 (2–3.7)	≤1.4	90.3
Astrup [10]	PCC	1500	3.06 (2.17–5.21)	≤1.5	74.3
Bitonti [14]	PCC	1500 units ^a^	4.58 ^b^	≤1.4	75
Gilbert [15]	PCC	1500 units: ICH ^c^ 1000 units: non-ICH ^c^	2.95 (2.2–3.0)	≤1.5	86.7

aPCC: activated four factor prothrombin complex concentrate. PCC: non-activated four factor prothrombin complex concentrate. Values are reported as means (standard deviations) or medians (interquartile ranges). ^a^ 2000 units if INR > 7.5 or >100 kg. ^b^ authors did not provide the standard deviation. ^c^ optional additional 500 units if INR > 10 or >100 kg.

## Data Availability

Not applicable.

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
