# Peer review of "Fixed-Dose Factor Eight Inhibitor Bypassing Activity (FEIBA) in the Management of Warfarin-Associated Coagulopathies"

_pharmacy, 2022, doi:10.3390/pharmacy10030050_

Round 1
Reviewer 1 Report
This is an interesting paper although the number of patients included in the study is small.
It can be published after some minor language check.
I suggest improving the "Conclusions" section in order to reinforce it.
Author Response
- Point 1: This is an interesting paper although the number of patients included in the study is small.
- Response 1: thank you for your comment.
- Point 2: It can be published after some minor language check.
- Response 2: manuscript reviewed and changes made where applicable.
- Point 3: I suggest improving the "Conclusions" section in order to reinforce it.
- Response 3: thank you for your comment. The conclusion section revised accordingly.
Reviewer 2 Report
Overall, this study was very interesting which has evaluated the Fixed dose FEIBA in the management of warfarin associated coagulopathies. Though it would be surplus to have larger number of patients, I also acknowledge that the human trials often do not go as planned. Just the part that made me to scratch my had was use of Mann Whitney U test, the stat used for nonparemetric variable. Other than that, the manuscript was well-written and easy to follow. I hope your institution could propose better FEIBA regimen based on findings from this study.

Author Response
- Point 1: Overall, this study was very interesting which has evaluated the Fixed dose FEIBA in the management of warfarin associated coagulopathies. Though it would be surplus to have larger number of patients, I also acknowledge that the human trials often do not go as planned. Just the part that made me to scratch my had was use of Mann Whitney U test, the stat used for nonparemetric variable. Other than that, the manuscript was well-written and easy to follow. I hope your institution could propose better FEIBA regimen based on findings from this study.
- Response 1: thank you for your comments. We used the Mann Whitney U test because our data was not normally distributed. This was a recommendation from our local statistician. We do aim to revise our protocol with our findings.
- Point 2: Please spell out FEIBA. (e.g., Factor VIII inhibitor bypass activity (FEIBA))
- Response 2: spelled out
- Point 3: For this study, did you get IRB exempt or obtained quality improvement project letter? While I know it is a retrospective study, usually it is still recommended to obtain IRB exempt or letter to certify this project being part of quality improvement project.
- Response 3: exempt
- Point 4: I would recommend to add the following sentence: "The total number of patients required additional dose can be found in the result section." In that way, the readers know where to find this information.
- Response 4: our study evaluated the effectiveness of a single FEIBA dose and did not collect data on the number of patients who received more than one dose.
- Point 5: This is a bit confusing. What was the reasoning behind using Mann-Whitney U test for continuous data? From my understanding, Mann-Whitney U test is used for nonparametric data. So I am questioning the reason behind using Mann-Whitney instead of student t-test that is specifically used for continuous variable. (https://www.statisticshowto.com/mann-whitney-u-test/; http://surgicalcriticalcare.net/Statistics/continuous.pdf). Please provide the reasoning or I would recommend to conduct stat analysis using student t test or other stat test for continuous variable.
- Response 5: We used the Mann Whitney U test because our data was not normally distributed. This was a recommendation from our local statistician.
- Point 6: Put comma here
- Response 6: added
Reviewer 3 Report
The authors investigated the institutions’ practice of administering FEIBA. They suggested that it was necessary to revise the current procedure of FEIBA. The approach and the data are interesting, and the manuscript is written well. However, there are some points required the revision. The detailed information was shown below.
- It is considered that the effect of FEIBA is dependent on not only the dose but also the patient background. Although authors summarized the patient background in Table 1, please consider adding the additional clinical information like liver function, renal function, hemoglobin and some risk factors. The information will be useful to interpret the data.
- The above clinical information may be related to the post-hoc analysis of patients in Table 3. Please consider that there are any tendency for clinical information between two groups. If authors can find any tendencies, please describe in the manuscript.
Author Response
- Point 1: It is considered that the effect of FEIBA is dependent on not only the dose but also the patient background. Although authors summarized the patient background in Table 1, please consider adding the additional clinical information like liver function, renal function, hemoglobin and some risk factors. The information will be useful to interpret the data.
- Response 1: additional info added to Table 1.
- The above clinical information may be related to the post-hoc analysis of patients in Table 3. Please consider that there are any tendency for clinical information between two groups. If authors can find any tendencies, please describe in the manuscript.
- Response 2: no tendencies were found.